# The Levels of Ghrelin, Glucagon, Visfatin and Glp-1 Are Decreased in the Peritoneal Fluid of Women with Endometriosis along with the Increased Expression of the CD10 Protease by the Macrophages

**DOI:** 10.3390/ijms231810361

**Published:** 2022-09-08

**Authors:** Aleksey M. Krasnyi, Alsu A. Sadekova, Tatyana Y. Smolnova, Vyacheslav V. Chursin, Natalya A. Buralkina, Vladimir D. Chuprynin, Ekaterina Yarotskaya, Stanislav V. Pavlovich, Gennadiy T. Sukhikh

**Affiliations:** 1National Medical Research Center for Obstetrics, Gynecology and Perinatology of Ministry of Healthcare of Russian Federation, Ac. Oparina Str. 4, 117997 Moscow, Russia; 2Department of Obstetrics, Gynecology, Perinatology and Reproductology of the Institute of Professional Education, I.M. Sechenov First Moscow State Medical University, Ministry of Healthcare of Russian Federation, B. Pirogovskaya Str. 2-4, 119991 Moscow, Russia

**Keywords:** endometriosis, peritoneal fluid, macrophages, ghrelin, GLP-1, glucagon, visfatin, CD10, CD86, BMI, energy metabolism

## Abstract

The aim of this study was to evaluate the levels of ten energy metabolism factors: C-peptide, ghrelin, GIP, GLP-1, glucagon, insulin, leptin, PAI-1 (total), resistin, and visfatin, and to determine the expression of GLP1R receptors, CD10, CD26 proteases, and pro-inflammatory marker CD86 by macrophages in the peritoneal fluid (PF) in patients with endometriosis. The study included 54 women with endometriosis and a control group of 30 women with uterine myoma without signs of endometriosis. The levels of factors in PF were assessed by a multiplex method. Expression of GLP1R receptors, CD10, CD26 proteases, and CD86 by macrophages was evaluated using flow cytometry. It was found that in women with endometriosis, the concentrations of ghrelin, GLP-1, glucagon, and visfatin in PF were reduced (*p* = 0.007, *p* = 0.009, *p* = 0.002, *p* = 0.008, respectively). At the same time, there was a noted increase in the CD10 protease expression by peritoneal macrophages (*p* = 0.044). Correlation analysis showed a positive correlation of ghrelin and GLP-1 levels with CD86 macrophage expression (*p* = 0.044, *p* = 0.022, respectively) in the study group; a positive correlation was also found between the levels of GLP-1, glucagon, and visfatin with CD26 macrophage expression (*p* = 0.041, *p* = 0.048, *p* = 0.015, respectively) in PF. No correlations were found in the control group. These results indicate that a decrease in the levels of ghrelin, GLP-1, glucagon, and visfatin in PF may contribute to endometriosis development through their impact on the expression of pro-inflammatory markers of PF macrophages.

## 1. Introduction

Endometriosis is a disease characterized by the presence of endometrial glands and stroma outside the uterine cavity. The prevalence of endometriosis in women of reproductive age is 10–15% [1]. Many theories try to explain the development of endometriosis, with Sampson’s theory of retrograde menstruation being the most common; however, the pathogenesis of the disease is still unclear [2]. One of the features of endometriosis is that the disease is associated with low adipose tissue content in the body composition and low waist-to-hip ratio [3]. It was shown that the association between abnormal adipose tissue homeostasis and endometriosis is clinically and biologically significant [4]. A review by Viganò et al. (2012) showed that BMI is reduced in women with endometriosis [5]. A possibility of non-hormonal therapy of endometriosis based on the regulation of energy metabolism is being studied [6,7]. However, the factors of energy metabolism in PF and their role in the development of endometriosis remain underestimated. Some publications report an increase in the peritoneal fluid leptin levels in endometriotic patients [8]. Studies of ghrelin concentration, in contrast, showed controversial findings [9,10].

It is known that macrophages are involved in adipogenesis and distribution of adipose tissue. It is assumed that thinness in women may be associated with a polarization of macrophages towards an anti-inflammatory profile [11]. Some authors believe that the development of endometriosis is also promoted by the polarization of peritoneal macrophages towards the M2 phenotype [12]. Our studies of peritoneal macrophages showed that even though the ratio of pro-inflammatory/anti-inflammatory markers decreased with increasing severity of endometriosis, there was no trend of polarization of macrophages to M1 or M2 states. There was a direct correlation between pro-and anti-inflammatory surface markers and cytokines in PF [13]. This finding indicates that the anti-inflammatory activity of peritoneal macrophages is reduced in endometriosis.

It has been demonstrated that some factors which regulate energy metabolism are expressed by macrophages and interact with the receptors on their surface; this may lead to changes in their pro-inflammatory and anti-inflammatory activity. For instance, in vitro experiments showed that leptin had an anti-inflammatory effect towards peritoneal macrophages of rats [14]. Some data demonstrate that GLP-1 induces polarization of macrophages, derived from monocytes, towards the M2 phenotype [15]. Ghrelin also regulates PF macrophages of mice [16]. The peritoneal macrophages are able to change the content of some factors in PF by affecting them with CD10 and CD26 membrane proteases. These proteases are known to degrade glucagon, GLP-1, and GIP.

The aim of this work was to study the levels and identify the relationships of the following factors in PF of patients with endometriosis: ten energy metabolism markers (C-peptide, ghrelin, GIP, GLP-1, glucagon, insulin, leptin, PAI-1 (total), resistin, and visfatin); the expression of GLP1R receptors and proteases CD10 and CD26 by macrophages, which can damage the mentioned signaling molecules; and the expression of surface pro-inflammatory marker CD86. In this study, we used the CD86 macrophage expression level as a pro-inflammatory marker. The expression of CD86 is typical for the M1 polarization state. Functionally, the expression of CD86 is necessary for macrophages to activate the cytotoxic activity of lymphocytes when interacting with the CD28 receptor on the lymphocyte membrane. We believe that the levels of pro-inflammatory markers of peritoneal macrophages are an important factor in the body’s fight against endometriosis. Further investigation of possible mechanisms of regulation of pro-inflammatory markers may give a deeper insight into the pathogenesis of the disease.

## 2. Results

Characteristics of patients with endometriosis and women of the control group are presented in Table 1. The patients did not significantly differ by age and menstrual cycle phase. At the same time, body mass index (BMI) was significantly lower in the group of women with endometriosis. In patients of the main group, stages 3–4 of endometriosis prevailed (63.2%). The ENZIAN classification was used to specify the localization and the size of the endometrioid foci, and the severity of uterus damage. Deep infiltrating endometriosis was noted in compartment A (rectovaginal septum, vagina) in 6 of 54 women (11.1%); in compartment B (uterosacral ligament, pelvic peritoneum) in 26 of 54 women (48.1%); and in compartment C (large intestine) in 37 of 54 women (68.5%). In the study group, patients with adenomyosis (FA) accounted for 42.5% of cases (23 of 54 women); patients with intestinal endometriosis (F1) were 61.1% (33 of 54 women), patients with ureteral endometriosis (FU) were 11.1% (6 of 54 women); and lesions of other localizations (anterior abdominal wall, FO) were present in 1.8% (1 of 54 women).

The expression of GLP1R, CD10, CD26, and CD86 by macrophages, as well as the levels of ten factors of energy metabolism in PF are shown in Figure 1. It was found that in women with endometriosis, the levels of ghrelin, GLP-1, glucagon, and visfatin in PF were decreased ((*p* = 0.007, *p* = 0.009, *p* = 0.002, *p* = 0.008, respectively). At the same time, an increase in CD10 protease expression by peritoneal macrophages was observed (*p* = 0.044). None of the studied parameters depended on the severity of endometriosis, and there were no statistically significant differences in the factors with respect to the menstrual cycle phase in either the control or the main group.

Correlation analysis (Figure 2) showed a positive correlation of ghrelin and GLP-1 concentrations with macrophage expression of pro-inflammatory marker CD86 (*p* = 0.044, *p* = 0.022, respectively) in the study group. A positive correlation was also found between the levels of GLP-1, glucagon, and visfatin in PF and CD26 macrophage expression (*p* = 0.041, *p* = 0.048 *p* = 0.015, respectively). No correlations were found in the control group. In addition, at the same time, there was a positive correlation between BMI and macrophage expression of all studied factors (*p* < 0.05) in the control group.

## 3. Discussion

The relationship between the immune system and energy metabolism is an attractive subject for research, mainly due to observations that inflammation is induced by an unhealthy diet and obesity [17,18]. Endometriosis is characterized by a low overall adipose tissue content, regarding female human body composition, and a lower BMI. At the same time, the state of the immune system in patients with low BMI has not been thoroughly studied. 

It is known that hormones associated with energy metabolism can directly affect the activity of macrophages by binding to the corresponding receptors on their surface. This may be crucial for the development of endometriosis, since it is believed that the PF macrophages eliminate the endometrial cells that are brought to the abdominal cavity by retrograde menstruation. Earlier, we studied pro- and anti-inflammatory activities of macrophages in the PF of endometriosis patients and came to the conclusion that in severe endometriosis both activities did not differ from those in the control group but were increased in patients with mild endometriosis [13]. These findings indicated that macrophages may not “recognize” severe endometriosis, possibly being under modulating influence of unknown factors. In this paper, we considered the potential factors that can affect macrophages. Since endometriosis is associated with overall low content of adipose tissue and changes in its distribution, factors of energy metabolism could act as signaling molecules which modulate macrophages. In previously published studies, an increase in PF leptin concentration was noted [19]. This finding was not confirmed in our study. At the same time, we found that leptin in both groups had a direct correlation with BMI (Figure 2). Rathore et al. showed that the concentration of ghrelin in PF of patients with endometriosis was significantly lower than in the control group [9]. Dziunycz et al. obtained opposite results [10]. Based on the data on increased leptin and ghrelin levels in the PF, Pantelis et al. suggested that endometriosis is associated with obesity, despite an obvious decrease in BMI in patients with endometriosis [20].

Our study did not reveal any contradictions between the decrease of adipose tissue in patients with endometriosis and the levels of energy metabolism factors in PF. There was a decrease in PF concentrations of signaling molecules associated with obesity, namely glucagon, GLP-1, visfatin, and ghrelin. It is known that the level of GLP-1 in plasma correlates with the extent of obesity [21]. In addition, plasma glucagon concentrations positively correlate with BMI [22]. The expression of visfatin is increased in patients with abdominal obesity and type 2 diabetes [23]. The level of ghrelin in PF seems to have no pronounced correlation with BMI; however, a positive correlation between ghrelin gene expression in gastric ghrelin cells, glycemic levels, and BMI in obese patients has been demonstrated [24]. To determine a possible relationship between the PF macrophages and selected factors of energy metabolism, we studied the expression of a pro-inflammatory marker, CD86, and proteases CD10 (Neprilysin) and CD26 (Dipeptidyl peptidase-4) of the GLP1R receptor by macrophages in PF.

It is known that macrophages make up 70–90% of the entire population of peritoneal leukocytes and play a significant role in the elimination of endometrial cells after retrograde menstruation. Therefore, the search for factors that can affect the macrophages cytotoxicity may be important for understanding the mechanisms of endometriosis development and elimination. CD86 is one of the macrophagal pro-inflammatory markers. It has an ability to bind, as a ligand, to a costimulatory molecule on a lymphocyte’s surface and activate it. Correlation analysis showed that in patients with endometriosis, the expression of CD86 by macrophages correlated with the levels of GLP1 and ghrelin in PF; these indicators were decreased compared with the control group. At the same time, BMI in the main group did not directly correlate with the expression of CD86 by macrophages. These findings show that there is no direct association between BMI and endometriosis development; nevertheless, certain energy metabolism factors may play important roles in the disease pathogenesis. No correlation between the levels of the energy metabolism and CD86 expression by macrophages was found in the control group. At the same time, BMI directly correlated with the expression of CD86 and other macrophage-expressed molecules studied in this work. This means that energy metabolism was strongly associated with the activity of PF macrophages in both groups. At the same time, we observed a difference in the mentioned correlation between the studied groups. CD10 and CD26, expressed by macrophages, are non-selective proteases that damage glucagon and GLP1. The role of proteases in the development of endometriosis is considered in regards to the invasion of endometrial cells into the peritoneum or an ovary [25]. In this study, we assessed the expression of proteases by macrophages as a potential factor of regulation of signaling molecules in the PF. We found a significant increase in CD10 expression in endometriosis, which possibly explains the decrease in glucagon, GLP-1, visfatin, and ghrelin. Direct correlation between the PF levels of the studied factors and macrophage expression of CD26 in the study group may indicate the presence of a positive inverse relationship, which leads to an increase in protease expression by macrophages, preventing signaling molecules of PF from reaching physiological level. We did not find any difference in the expression of GLP1R by macrophages between the control and the study group. In this study, we found new correlations between the factors of energy metabolism and the pro-inflammatory markers of PF macrophages. These results may serve as a background for further in vitro and in vivo studies in animal models, which may reveal unknown mechanisms of endometriosis development. 

## 4. Materials and Methods

### 4.1. Ethics and Sample Collection

This study project was approved by the Ethics Committee of the National Medical Research Center for Obstetrics, Gynecology, and Perinatology. The patients signed an informed consent for study of their biologic material. All patients were operated in the Department of Surgery of National Medical Research Center for Obstetrics, Gynecology, and Perinatology. 84 women were included in the study. 

The patients were divided into two groups. Group 1 included 54 patients diagnosed with endometriosis in various combinations: retrocervical endometriosis, peritoneal endometriosis, and ovarian endometriosis. In all patients of the main group, preoperative pelvic ultrasound examination was performed, and MRI (if indicated). This allowed the exclusion of any other pelvic mass concomitant to endometriosis.

Endometriosis was staged according to the American Society for Reproductive Medicine (ASRM) classification and ENZIAN-score [26,27]. Group 2 (control group) included 30 patients with uterine myoma with no signs of endometriosis at laparoscopy. Patients with uterine myoma were chosen as the control group because they needed surgery and therefore could provide peritoneal fluid samples, non-affected by the disease, since uterine fibroids are benign incapsulated neoplasms. Samples of the PF were obtained from the patients during laparoscopy prior to any surgical manipulation. Patients with diabetes mellitus were excluded from the study.

### 4.2. Concentration of Energy Metabolism Markers

Samples of the PF were centrifuged at 3000× *g* for 10 min and stored at −80 °C. Determination of the levels of energy metabolism markers: C-peptide, ghrelin, GIP, GLP-1, glucagon, insulin, leptin, PAI-1 (total), resistin, and visfatin in PF was performed using a standard 10-plex test system Bio-Plex Pro Human Diabetes Panel, (Bio-Rad, Hercules, CA, USA) on a Bio-Plex 200 System (Bio-Rad, Hercules, CA, USA). The results were processed using Bio-Plex Manager 6.0 Properties application (Bio-Rad, Hercules, CA, USA). The concentration of cytokines is presented in pg/mL.

### 4.3. Flow Cytometric Analysis of Expression of CD Antigens

Macrophages were isolated from PF within 1 hour after obtaining the samples. Monoclonal antibodies CD86, GLP1R, CD10, and CD26 (Biolegend, San Diego, CA, USA) were used for immunophenotyping. A macrophage gate was identified basing on side scatter (SSC) and CD45 expression (BD Biosciences Systems & Reagents, San Jose, CA, USA). BD FacsCalibur flow cytometer with CellQuest Pro software (BD Biosciences Systems & Reagents, San Jose, CA, USA) was used for analysis. To assess the expression, median fluorescence (MFI) was determined for each antibody in each sample.

### 4.4. Statistical Analysis

The significance of the differences between the groups was analyzed using the two-sided Mann–Whitney U test. The results are presented as the median, upper, and lower quartiles (Me (Q1; Q3)). The Spearman’s correlation coefficient (r_s_) was applied to study the relationships between the studied factors. Categorical variables were analyzed using a Fisher’s exact test. Differences were considered statistically significant at *p* < 0.05. SPSS Statistics 17 (IBM, Armonk, NY, USA) and OriginPro 8.5 (OriginLab Corporation, Northampton, MA, USA) programs were used for the statistical processing of the results and plotting graphs.

## 5. Conclusions

In our study, we showed that BMI is not directly associated with endometriosis; however, certain factors of energy metabolism may play an important role in the development of the disease, through their impact on the pro-inflammatory properties of macrophages. We believe that further study of PF energy metabolism factors and their impact on macrophage activity may contribute to the understanding of the pathogenesis of endometriosis.

## Figures and Tables

**Figure 1 ijms-23-10361-f001:**
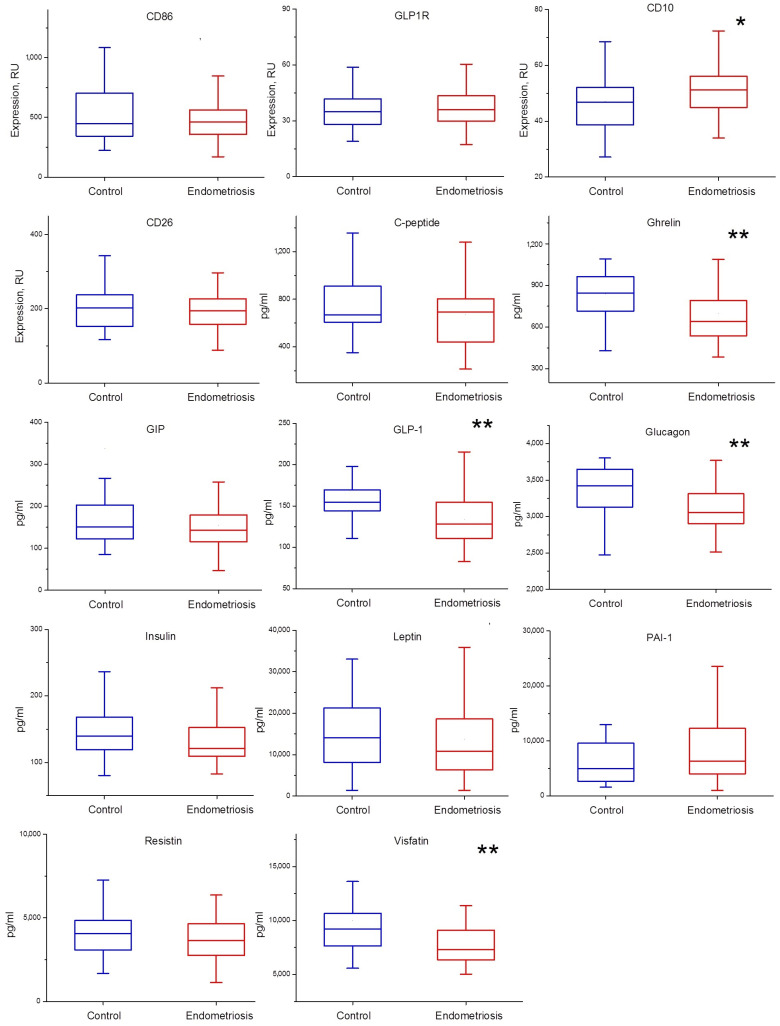
Concentrations of energy metabolism factors in the PF in control and endometriosis groups, *—*p* < 0.05, **—*p* < 0.01.

**Figure 2 ijms-23-10361-f002:**
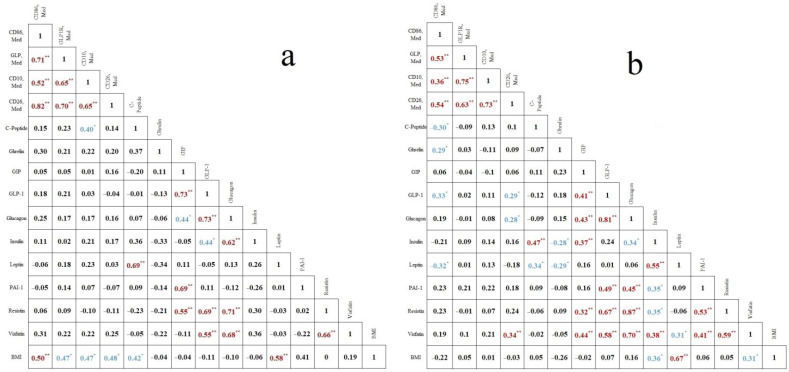
Correlations between energy metabolism markers (C-peptide, ghrelin, GIP, GLP-1, glucagon, insulin, leptin, PAI-1 (total), resistin, visfatin), GLP1R receptors, CD10 and CD26 proteases, and pro-inflammatory marker CD86. (**a**)–control group, (**b**)–endometriosis group, *—*p* < 0.05 (blue), **—*p* < 0.01 (red).

**Table 1 ijms-23-10361-t001:** The clinical characteristics of the study groups.

		Endometriosis n = 54	Control n = 30	*p*-Value
Age (year)		35 (31; 37.75)	37 (31.75; 40)	0.11
BMI (kg/m^2^)		21.3 (20.5; 24.2)	24.9 (22.6; 27.8)	0.05
Menstrual cycle phase	Proliferative	18 (33.3%)	11(36.7%)	0.81
	Secretory	23 (42.6%)	14 (46.7%)	0.82
	Menstrual cycle disorder	13 (24.1%)	5 (13.3%)	0.58
Stage of endometriosis	Stage I–II	20 (37.03%)		
	Stage III–IV	34 (62.97%)		
Type of myomas	Intramural-subserous myoma		19 (63.3%)	
	intramural-submucous myoma		11 (36.7%)	

Values are presented as median Me (Q1;Q3) or as number (%).

## Data Availability

The data that support the findings of this study are available from the corresponding author, upon request.

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
