# Peer review of "The Levels of Ghrelin, Glucagon, Visfatin and Glp-1 Are Decreased in the Peritoneal Fluid of Women with Endometriosis along with the Increased Expression of the CD10 Protease by the Macrophages"

_ijms, 2022, doi:10.3390/ijms231810361_

Round 1
Reviewer 1 Report
Dear authors, thank you for your paper which I read with a lot of interest.Here are my comments:
- Material and methods: Why did you choose patients with myomas as a control group? These are both estrogen-dependent diseases.
How was it possible to ensure, with a preoperative informed consent, that none of the patients in the endometriosis group had myomas? and vice-versa? Do you think that this control-group is really ideal?
You tested for energy metabolism factors with are associated with adipose tissue. The BMI of your two study groups however significantly differed - This creates a possible bias. Perhaps your results merely reflect the difference in BMI as opposed to a difference between the two diseases.
How was the peritoneal fluid collected? Was it pure PF or was it lavage? Had any patients undergone hysteroscopy previous to laparoscopy?
You stated that you used the ENZIAN classification - this score should then be included in the patient-characteristics table. Please also see citation here (is 23 correct for the ENZIAN?? - where is citation 24)
Author Response
Dear reviewer, thank you for your comments, we will gladly answer your questions.
- The best choice of the control group would be healthy women, but for obvious reasons this is not possible. Myoma is a benign incapsulated neoplasm that does not directly contact the peritoneal fluid, which makes direct interaction between PF macrophages and the tissue of the neoplasm significantly complicated, and therefore, has considerably less impact on the PF than endometriotic cells entering the abdominal/pelvic cavity with retrograde menstruation or coming from endometriosis foci. We believe that estrogen has no significant effect on the studied factors, since we did not find any differences between groups with regard to the menstrual cycle. We did not exclude patients with both endometriosis and myoma for the reasons mentioned above. We referred these patients to the study group.
- In our study, we found that the energy metabolism factors that had demonstrated the greatest differences between the study group and the control group were not directly correlated with BMI. This fact is supported by the presented table with correlation coefficients. The only factor associated with BMI was leptin, but it did not differ significantly between the groups.
- As noted in Materials and Methods section, PF samples were obtained from the patients during laparoscopy prior to surgical manipulation. Exclusively for this study, we agreed with the surgeons on a surgical protocol, which included PF sampling prior to hysteroscopy. Patients whose PF could not be collected due to its small amount were excluded from the study.
- The severity of endometriosis was assessed according to ASRM. We also added a paragraph with the endometriosis foci localization according to ENZIAN.

Reviewer 2 Report
I read the presented manuscript with interest. Unfortunately, the work shows differences to the control group, which differs significantly in age and BMI. Additionally, it describes correlations suggesting a cause-and-effect relationship, which is not proven.
-
In abstract: drawing functional implications based solely on correlations, found all the more so only in the endometriosis group, seems to be speculation. In addition, the authors use the level of CD68 macrophages interchangeably with the activity of macrophages.
-
A large number of authors and the limited contributions of some of them suggest that it may be more of honorary authorship. Please follow the COPE policy on authorship, contributorship and acknowledgment.
-
The group of case and control patients differed significantly in terms of age (p at the borderline of statistical significance) and BMI, which in the case of the assessment of markers related to metabolic balance is of key importance. Therefore, drawing further conclusions seems to be pointless.
Author Response
Dear reviewer, thank you for your comments, we have tried to give meaningful responses to them.
- In this study, we used the CD86 macrophage expression as an indicator of pro-inflammatory activity of macrophages. CD86 expression is typical for the M1 polarization state. Functionally, CD86 expression is necessary for macrophages to activate the cytotoxic activity of lymphocytes when interacting with the CD28 receptor on the lymphocyte membrane. One of the objectives of this study was to find a correlation between the studied factors and the pro-inflammatory activity of macrophages in endometriosis marked by CD86 expression. The noted correlations cannot be claimed as certain, but these findings open a perspective for further in vitro or in vivo studies in animal models.
- We have included a number of surgeons in the authors list. They, at our request, have changed the surgical protocol and collected a pure PF. Each of the co-authors made a significant contribution to the study, and all authors are responsible for its results.
- A p-value of 0.06 indicates that the groups had no significant differences by age. When we had excluded the two youngest patients from the study group, the p-value increased to 0.1, and this did not affect the results. However, we did not change our data for the reason mentioned above. We expected a lower BMI in the women in the study group, moreover, the literature data on lower BMI in endometriosis was one of the backgrounds for this study. We added the link to the source in the introduction.
Reviewer 3 Report
They reported that metabolism factors and macrophage in peritoneal fluid in patients with endometriosis. Their experiments are just simple, and if they don't discuss them well in the introduction and discussion, we won't know what this paper is about.
Major points;
1. It is easy to imagine that the factors are different for patients with endometriosis because their BMI is lower than that of normal individuals. Therefore, it is important to understand how these factors are related to macrophages and to endometriosis. Their explanation seems inadequate in Introduction.
2. Recently macrophages have been discussed separately as M1 and M2, but do M1 and M2 differ in their involvement in metabolism factors?
3. The following paper seems important to discuss your thesis, but I could not read it because it is written in Russian.
[12] Krasnyi, A.M. et al. The content of cytokines IL-6, IL-8, TNF-α, IL-4 and the level of CD86 and CD163 expression in peritoneal fluid 265 macrophages has a reverse correlation with the degree of severity of external genital endometriosis.
4. Did they intend that gaining weight would cure endometriosis?
Minor points;
1. Are metabolic abnormalities (diabetes, lipid disorders, etc.) excluded from the patients?
2. The letters in Figure 2 are small and appear blurred when enlarged.
Author Response
Dear reviewer, thank you for your comments, we tried our best to make meaningful responses to them.
- We added more background to the study, as well as some literature data in the introduction; also we specified the purpose of the study. The available literature data on the impact of metabolism factors on PF macrophages is scarce and contradictory. We mentioned the most interesting ones in the Introduction.
- We have changed DOI to English version in the reference to the article by Krasnyi, A.M. et al.
- Based on the study mentioned above, we can state that regarding PF macrophages, it is more correct to speak not about individual states of M1 and M2 polarization, but about reduced pro-inflammatory and anti-inflammatory activity of macrophages in endometriosis.
- In our study, we focus on pro-inflammatory (cytotoxic) activity of PF macrophages and search for possible associated factors.
- In the study group, we did not found any correlation between CD86 macrophage expression (pro-inflammatory activity) and BMI. This indicates that weight gain will not induce the macrophages to cure endometriosis more effectively. It is interesting that in the control group such a correlation exists, maybe that is why they do not have endometriosis.

Round 2
Reviewer 1 Report
How did you ensure that leiomyoma were not present in endometriosis patients?
Author Response
Dear Reviewer, thank you for your valuable comments, which helped us to make our article better. Regarding leiomyoma in the control group, since all patients underwent ultrasound examination, and most of them also underwent MRI for invasion of endometriosis, we had all the information about their pelvic structures and lesions. We have added this information to our article.
Reviewer 2 Report
The article was not revised. I have read the authors' answers, but they do not convince me. If we are talking about membrane expression, let's not call it activity. If the control group differs from the test group, we should select the appropriate control group. Please follow the COPE policy on authorship, contributorship, and acknowledgment. If some people are involved in taking samples or operating on patients - certainly they can be acknowledged (but not autors!), because this is not what scientific work is about. I do not add my nurse as the author because she takes blood from patients! The test requires major changes and may be considered after resubmission.
Author Response
Dear Reviewer, thank you for your valuable comments, which helped us to make our article better. We have tried to take all your comments into account. We excluded the three youngest patients from the study group so that the p-value for age changed from “p < 0.1 weak evidence or a trend” to “p ≥ 0.1 insufficient evidence”. This did not change the results dramatically, however the differences between the groups became more pronounced. We also changed the term “pro-inflammatory activity”, used to describe CD86 expression, to commonly used term “pro-inflammatory marker”. We have also revised the number of authors and moved those who did only technical work to the Acknowledgements section.
Reviewer 3 Report
- The author have not responded to my points.
- Figure 2 aren't improved.
- I am not asking if patients have diabetes. I am asking whether BMI or metabolic factors are associated with endometriosis as the conclusion of this paper.
Author Response
Dear Reviewer, thank you for your valuable comments, which helped us to make our article better. We have edited the conclusion as you requested. This really improved the article and made it more complete. We also made the resolution of Figure 2 higher. We edited the Introduction after the first round of peer review. Regarding your question about M1 and M2 polarization and its correlation with metabolic factors from the first round of peer review, we have not found any data about human peritoneal macrophages. Blood-derived macrophages are known to acquire an M2 state phenotype under the influence of GLP1, but a similar effect on peritoneal macrophages could not be expected. We have previously shown that peritoneal macrophages do not polarize toward the M1 or M2 state, and the expression levels of M1 and M2 markers change codirectionally. Therefore, it was not possible to predict the result based on the literature data prior to the experiment. I hope I have clarified our point stated in the Introduction.
Round 3
Reviewer 3 Report
Your article is improved.